# A Study of the Ping An Health App Based on User Reviews with Sentiment Analysis

**DOI:** 10.3390/ijerph20021591

**Published:** 2023-01-16

**Authors:** Fang Fang, Yin Zhou, Shi Ying, Zhijuan Li

**Affiliations:** 1School of Information Management, Wuhan University, Wuhan 430072, China; 2School of Computer Science, Wuhan University, Wuhan 430072, China; 3Department of Economics and Management, Wuhan University, Wuhan 430072, China

**Keywords:** m-Health apps, topic model, user reviews, dimension mining, dimension weight, sentiment analysis

## Abstract

By mining the dimensional sentiment and dimension weight of the Ping An Health App reviews, this paper explores the changing trend of the influence of dimensions on user satisfaction and provides suggestions for the Ping An Health App operators to improve user satisfaction. Firstly, the topic model is used to identify the topic of user comments, and then the fine-grained sentiment analysis method is used to calculate the sentiment and weight of each dimension. Finally, the changing trend of the weight of each dimension and the changing trend of user satisfaction of each dimension are drawn. Based on the reviews of the Ping An Health App in the Apple App Store, users’ concerns about Ping An Health App can be summarized into seven main dimensions: Usage, Bug report, Reliability, Feature information, Services, Other apps, and User Background. The “Feature information” dimension and “Reliability” dimension have a great impact on user satisfaction with the Ping An Health App, while the “Bug report” dimension has the lowest user satisfaction.

## 1. Introduction

With the development of information and communication technology, mobile health has become a trend. Mobile health, or m-health, is a promising tool for supporting healthcare, using mobile technologies and innovative applications to solve health problems that can provide information and resources to healthcare professionals and patients [1]. With the rise of Web 2.0 and Web 3.0, the emergence of smart mobile devices that support 3G and 4G mobile networks provide unprecedented conditions for the development of m-Health apps [2,3,4,5]. M-Health apps were downloaded more than 200 million times in 2010, and 70% of smartphone users visited at least one mobile health app, the number of downloads of medical apps is expected to reach 305 million by 2022, m-Health apps are convenient for providing healthcare at any time and place [5,6,7].

Mobile consultation is an important channel to improve the quality of medical services, which can alleviate the uneven distribution of medical resources and improve the efficiency of users in receiving medical services [8,9,10]. More and more researchers devote themselves to m-Health app research to maximize the role of m-Health apps and make it easier for patients to obtain medical services. At present, the research on m-health apps mainly focuses on use intention and influencing factors [11,12,13], m-health app development [14,15,16,17,18,19,20,21], controlled trial studies on the effectiveness of m-health apps [22,23,24,25,26,27,28], research on the use of specific apps [29,30].

### 1.1. m-Health Apps in China

Some Chinese scholars also devote themselves to m-Health app research. Xu Qian et al. investigated the top 100 m-Health apps in Apple’s online store and analyzed the public’s demand for health information, pointing out that users focus on the health information that is related to health promotion, healthcare, and consulting and services for health tracking that can be easily completed [11]. Based on the trust theory, Zhang Min conducted log tracking and questionnaire survey experiments to analyze factors influencing trust in mobile medical apps for medical inquiry from the perspective of multi-perspectives of process, institution, and characteristic. Zhang Min pointed out that when patients believe that the information on the app is relatively complete, reliable, authentic and timely, their trust in the app will be significantly improved [12]. Meng-Yan Tang et al. investigated ambulate surgery patients’ preferences for m-health apps and their intention to use them through questionnaires, and the results showed that ambulatory surgery patients were willing to use m-health apps. Then, the author designed the framework of an m-health app based on the research results, which includes four sections: My Ambulatory Surgery, Online Consultation, Postoperative Online Follow-up, and Health Information [13].

Most research on user satisfaction with mobile medical apps is conducted in the form of a questionnaire survey. In this paper, user review data from app stores are used. Different from the methods of unsupervised learning and sentiment dictionaries used in previous studies, deep learning and fine-grained sentiment analysis will be used in this study.

Aspect-level sentiment analysis is a popular field of sentiment analysis with important application value, which focuses on the mining of fine-grained sentiment information, and aims to infer the sentiment polarity of each aspect in the text. The goal is to extract the sentiment polarity for each aspect of an entity [31]. Short text sentiment analysis based on deep learning can be generally divided into five stages: text preprocessing, initial word vector representation, feature extraction, and contextual semantic representation, deep learning model training, and testing, experimental result analysis, and evaluation. Fine-grained sentiment analysis based on sequence labeling redefines the target/aspect level fine-grained sentiment tendency analysis task by combining the aspect information labeling task and the sentiment information labeling task [32].

### 1.2. Research on App Store Reviews

With the development of the Internet and the increasing number of mobile device users, people’s demand for various mobile applications is increasing. Since the launch of Apple’s App Store in July 2008, the number of app stores has grown. There are three main types of app stores, depending on the mobile device: Apple App Store for IOS, App Stores for Android (360 Mobile Assistant, Huawei App Market, Google Play, etc.), and Windows Store for Windows phones.

App refers to any software that can be installed by users who have a suitable platform, and the installation process does not require the help of professional and technical personnel [33]. In addition to providing users with downloads, the app market or app store also allows users to review the app in the form of ratings and reviews [34].

The number of m-Health app users is increasing, and some users are willing to give feedback on m-Health apps through the app store. User reviews may include user experience and functional requirements for apps, which can help software developers understand user attitudes and improve software quality. By mining users’ reviews, software developers can explore users’ functional requirements for the software. Hoon et al. published the first literature focusing on app store reviews in 2012 [33]. In the same year, Goul et al. conducted sentiment analysis on 5000 reviews of the Apple App Store to promote software requirements engineering [35]. Ha et al. manually examined 556 reviews of 59 Google Play apps and classified them into topics and sub-topics based on the content of the reviews. They found that most of the information in the reviews was about the quality of the app, rather than security or privacy issues [36]. Hoon et al. collected 29,182 reviews of the top 25 Health and Fitness apps from the Apple App Store. Hoon points out that these reviews are mostly composed of sentiment words and are closely related to the review’s star rating [37]. Ciurumelea et al. divided app reviews into eight categories through manual analysis [38]. Sriosopa et al. conducted an exploratory study on 7198 reviews of six Internet of Things (IoT) products. They used content analysis to refine 7198 reviews based on Ciurumeleal’s research and divided 7198 reviews into 8 high-level categories and 12 low-level categories [39].

App Store reviews are a bridge between users, developers, and software providers. Users review the app store to express their support for the application and to communicate with developers, software providers, and other users through reviews. Therefore, app store reviews contain a lot of useful information, such as whether users are satisfied with the app, expected improvements and potential needs, whether they are willing to recommend it to others, and even problems with the app. Analyzing user reviews helps to understand whether mobile medical programs can really meet people’s needs and whether users are really satisfied with these medical services provided by the medical industry, so as to put forward improvement suggestions for service providers, improve software and service quality, and enhance user satisfaction [34].

This study aims to explore the impact of the Ping An Health App on user satisfaction through user reviews. This study attempt to answer the following questions: (1) What are the user review topics of the Ping An Health App? What are the dimensions that users focus on? (2) What is the impact of each dimension on user satisfaction? To what extent are the dimensions contributing to user satisfaction? (3) What is the changing trend of the influence of each dimension and user satisfaction?

## 2. Materials and Methods

The purpose of analyzing m-Health app user reviews is to identify the topics and dimensions of m-Health app reviews, identify user’s attitudes towards dimensions of m-Health apps, calculate the influence of each dimension on user satisfaction, explore the changing trend of dimensional satisfaction, and then improve the software quality of m-Health apps. Based on text mining technology, this study calculates the sentiment and influence of each dimension: (1) crawling user review data through python and preprocessing the data; (2) identifying review topics based on the Contextual Topic Identification model, extracting the dimensions and topic words of user reviews, and expand the topic words based on the word vector model; (3) extracting dimension and dimensional sentiment based on fine-grained sentiment analysis; (4) analysis of changes in user satisfaction. The research framework is shown in Figure 1.

### 2.1. m-Health Apps User Review Data Acquisition and Processing

Because the Ping An Health App offers a global edition on the Apple App Store, this study uses the reviews of the Apple App Store. First, use Python to obtain the title, rating star, text, and time of each review, and then clean the data. It mainly includes removing advertising information, repeated reviews, and invalid reviews with too few words.

### 2.2. Analysis of User Reviews Based on Topic Model

First, identify the topics through the topic model, and then combine the functional characteristics of the m-Health app to obtain the categories and feature words of m-Health app reviews. Finally, based on the Word2vec word vector model, the feature words are expanded to obtain the main category feature word set of m-Health apps that reflects users’ concerns in online reviews.

The topic model maps high-dimensional word space to low-dimensional target topic space to achieve dimension reduction, information summary, and summary of target documents [40]. Blei proposed the three-level probabilistic topic model LDA (Latent Dirichlet allocation) in 2003, where all document parameters are linked through a probabilistic generative model, and the multinomial distribution of topics in each document is generated via a Dirichlet prior associated with all documents in the corpusBlei proposed the three-level probabilistic topic model LDA (Latent Dirichlet allocation) in 2003, where all document parameters are linked through a probabilistic generative model, and the multinomial distribution of topics in each document is generated via a Dirichlet prior associated with all documents in the corpus [41]. Steve designed a contextual topic identification model to identify meaningful topics for sparse Steam reviews [42]. Nicole Peinelt proposed a tBERT model combining topic models and BERT for semantic similarity comparison in specific fields [43]. This study first allocates vectors through the probability topic of LDA, then embeds vectors through BERT sentences, and finally connects the two vectors. Then, the collected corpus is used to train the Word2Vec word vector model, and based on this model, the evaluation dimension feature words involved in user reviews are expanded.

### 2.3. Extracting Dimension and Dimensional Sentiment

Different users use m-Health apps for different purposes, which leads to a difference in the focus of each review, so the attention to each dimension in each review will also be different. App product review data already have ratings, but each review may contain multiple dimensions. The user’s sentiment towards the product is related to the sentiment of each dimension and the weight of each dimension, which in turn is related to the probability of the dimension mentioned in the whole review and the consistency of the dimension of sentiment with the overall sentiment of the review [44].
(1)$$Weight_{d}=Ref_{d}\times Con_{d}$$

Therefore, this paper extracts the dimensions contained in each review, counts the dimensions contained in each review and the number of times each dimension appears, and then calculates the probability of each dimension being mentioned in all reviews and the consistency of sentiment of each dimension with the overall sentiment of online reviews, finally getting the weight of each dimension.

In this paper, a deep semantic sentiment analysis framework, PaddleNLP, is used to conduct fine-grained sentiment analysis on review text. PaddleNLP is an easy-to-use and powerful NLP library with an awesome pre-trained model zoo, supporting a wide range of NLP tasks from research to industrial applications [45]. PaddleNLP supports several pre-trained models, its sentiment analysis module is based on the SKEP pre-trained model, PP-MiniL pre-trained model, and quantized PP-MiniL pre-trained model [46]. The whole process of sentiment analysis roughly includes two stages, which are the review opinion extraction model and the attribute level sentiment classification model. For a given piece of text, we first extract the potential comment attribute and the corresponding opinion of the attribute in the text sentence based on the former, and then concatenate the comment attribute, opinion, and the original text, and pass it to the aspect-level sentiment classification model to identify the sentiment polarity of the review attribute. In this study, a part of the dimensions is identified by using the dimension feature words, and then the sentiment of each dimension is manually annotated. Then, the sentiment analysis module is invoked to train the model, and finally, the dimension and sentiment polarity contained in each review is output.

### 2.4. Analysis of Changes in User Satisfaction

Through topic mining and the calculation of dimension weight, we can obtain the dimensions that users pay attention to, as well as the changing trend of the weight of each dimension and the changing trend of user satisfaction with each dimension. These results can help m-Health apps obtain valuable information from the perspective of users, so as to better improve product functions and design to meet user needs.

## 3. Results

### 3.1. Data Acquisition and Processing

On 27 January 2021, Ping An Health Medical Technology Co., Ltd. announced that the Ping An Good Doctor APP, an online medical service platform, was officially renamed Ping An Health APP [47]. As the largest national online medical portal in China, the Ping An Health App is a widely used online medical service platform, aiming to build a one-stop, full-process, online-to-offline service platform, and an online comprehensive health service platform integrating private clinics, pharmacies, health check-up, testing centers, and other offline health services [48,49]. By the end of 2019, the Ping An Health App had 315 million registered users and 729,000 daily consultations for medical services throughout the year [50]. According to the Annual Report of China Mobile Medical Market from 2012 to 2017 released by IResearch, there are more than 3000 medical and health apps in China. By the end of 2015, the Ping An Health App ranked first in the field of mainstream mobile medical and health applications in China with a coverage rate of 25.5% [51]. This study used python to capture user reviews of the Ping An Health App in the Apple App Store from 2014 to June 2021, and a total of 17,072 reviews were collected. After data cleaning, 11,764 valid reviews were obtained.

### 3.2. Identification of User Review Topics

According to the research results, the number of topics in this experiment is estimated to be 5–7 [52]. The experimental results show that when the number of topics is 7, the consistency score is the highest, and the contour coefficient of the clustering effect is the highest. According to the method, the name of each topic is manually assigned by the author according to the topic, which is a common practice [53,54]. The topics identified in the reviews through topic mining are listed below. The seven topics are: Usage, Bug report, Reliability, Feature information, Services, Other apps, and User Background.
Usage: Software usability mainly involves whether the user interface is beautiful, whether the page style is acceptable, and whether the software is easy to use, etc.;Bug report: Software errors mainly include software errors during user use, such as stalling or flashback, compatibility problems between the software and the mobile phone model, problems encountered in downloading and installing the new version, etc.Reliability: Software reliability mainly includes users’ reviews on software security, personal privacy, physicians’ professional level, and whether the software frequently sends short messages to users;Feature information: The functional features of the software mainly refer to the user’s evaluation of specific functions, such as online registration and consultation, online shopping, health column, and weight loss columns, as well as some featured activities and cost issues;Services: Service mainly includes users’ evaluation of software customer service and physicians’ service;Other apps: Other apps refer to the comparison between the Ping An Health App and other apps, including the comparison between medical apps and shopping apps, and whether the user is willing to recommend the Ping An Health App to others;User Background: Users describe their identity and purpose in reviews.

Combine the above seven topics with the research of Ciurumelea and Srisophak to subdivide the focus dimension of user reviews [38,39]. Reviews of the Ping An Health App can be divided into 7 main dimensions and 17 sub-dimensions, as shown in Table 1.

### 3.3. Dimension of Sentiment Calculation

To improve the accuracy of the model, this study first randomly 3000 data using the annotation tool Doccano for manual annotation. Doccano, short for Document Annotation, is an open-source text annotation tool that can be used to mark corpora for NLP tasks. It supports sentiment analysis, named entity recognition, text summarization, and other tasks [55]. The annotated data is then used to train the model, and the feature dimensions and sentiment polarity of all reviews are finally obtained. The mentioned probability, sentiment attribute, sentiment consistency, and comprehensive weight results of each dimension are shown in Table 2. By observing the collected reviews, it is found that users pay different attention to each dimension of the Ping An Health App. The higher the attention level of the dimensions, the greater the impact on the overall sentiment of the online reviews. Therefore, the influence of different dimensions of the An Good Doctor App on user satisfaction is measured by calculating the weight of each dimension in user satisfaction evaluation.

In general, “Feature information” is the dimension most frequently mentioned by users, and its sentiment has the highest consistency with that of the overall reviews. In other words, for the Ping An Health App, “Feature information” has the greatest impact on user satisfaction. The weight of the “Usage” dimension is higher than that of the “Reliability”, “Other”, “Service”, “Bug report”, “Background” and “other” dimensions. From the reviews, users often mention the convenience, smoothness, and memory size of the app. It can be seen that “Usage” is also an important factor affecting user satisfaction. In addition, users pay relatively little attention to the “Bug report” and “Background” dimensions, which indicates that users have a high tolerance for the error rate of the Ping An Health App. To better understand whether the attributes that users care about have changed, the author conducted a year-by-year analysis of the data from 2016 to 2020, and the results are shown in Figure 2 and Table 3.

As can be seen from Table 3, since 2016, the most and least mentioned dimensions by users were “Feature information” and “Background”, which were consistent with the overall trend. The “Reliability” dimension has increased in weight since 2016 and stabilized in second place by 2018. This shows that nowadays when using the Ping An Health App, users pay more attention to whether their privacy can be protected and whether the physicians they meet are authoritative and reliable. Some users expressed that some physicians of the Ping An Health App are not as professional and reliable as hospitals, and questioned the qualifications of physicians. The weight of the “Service” dimension began to rise after 2018, ranking third in 2020. Among them, the weight of the “Bug report” rose to third in 2019 and down to sixth in 2020, which indicates that the tolerance degree of users to the “Bug report” dimension decreased in 2019. Another significant increase in dimension weight is the “Service” dimension, which indicates that users have increased requirements for services. According to the data from the last five years, these three dimensions have a significant impact on users’ negative emotions.

### 3.4. Analysis of Changes in User Satisfaction

In this paper, user satisfaction is calculated by user positive propensity probability, and the user satisfaction score of the Ping An Health App in each dimension is calculated according to the results of fine-grained sentiment analysis. The results are shown in Figure 3 and Table 4.

It can be seen from Figure 3 and Table 4 that regardless of the comprehensive satisfaction score in the past five years or the satisfaction score in each year, users have the highest satisfaction score for “Background” and the lowest satisfaction score for “Bug report”. Among the annual satisfaction scores, 2019 has the lowest user satisfaction score.

Table 5 is obtained by sorting the user satisfaction scores from 2016 to 2020. User satisfaction with the “Background” dimension and “Usage” dimension ranks high, while user satisfaction with the three dimensions of “Feature information”, “Reliability” and “Service” fluctuates somewhat, but is generally low.

## 4. Discussion

The objective of this research study is to identify the dimensions that users pay attention to in the comments of the Ping An Health App and the influence of dimensions on user satisfaction and determine the influence of each dimension and the changing trend of user satisfaction in each dimension. Based on the analysis results, the author proposed improvement suggestions for m-Health app service providers to improve the quality of software and services and enhance user satisfaction.

The “Feature information” dimension is a key factor affecting user satisfaction with the Ping An Health App, but user satisfaction with the “Feature information” dimension is low. It can be seen from Table 3 and Table 5 that the weight of the “Feature information” dimension is always higher than other dimensions, and the satisfaction score of the “Feature information” dimension ranks low, whether it is annualized data or comprehensive data. This indicates that the dimension of “Feature information” is the key factor affecting user satisfaction with the Ping An Health App, which is consistent with the conclusion of the research [56]. Therefore, this paper believes that the key to improving user satisfaction with the mobile medical platform is to improve the functions of the mobile medical platform. In the beginning, the Ping An Health App introduced free consultations and held many special events to attract users. Later, some users expressed dissatisfaction with the change of consulting fees from free to charge, and had opinions on the inability to redeem gifts for special activities such as “step by step to win gold”. Some users also have objections to the charging standard. Some comments mentioned that the time for charging questions was not enough. For example, the time of a user’s consultation was 15 min, and some doctors replied slowly so that the user did not ask all the questions within 15 min. Some users think that the charge is high, and when the cost of using the app is the same as the cost of going to the hospital, they are more inclined to go to the nearby hospital instead of using the Ping An Health App. The operator of the app can set different fees for users to choose from according to the doctor’s location, work unit, and doctor qualification.

Both the impact and satisfaction of the “Reliability” dimension are increasing. Deng pointed out that trust, service quality, functional value, and emotional value have a significant impact on user satisfaction [57]. Yao built a mobile medical user satisfaction model and conducted empirical analysis through questionnaires to prove that the safety experience and reliability experience have a significant impact on user satisfaction with the mobile medical platform [58]. The “Reliability” dimension has increased in weight since 2016 and stabilized in second place by 2018. The ”Reliability” dimension contains four subdimensions: Security, Privacy, Qualification of a physician, and News Feed. From the content of the review, users have certain concerns about whether their privacy is protected and whether the professionalism of doctors can be guaranteed. Therefore, this paper believes that improving the security of the mobile medical platform and demonstrating the professionalism of doctors is conducive to improving the satisfaction of users of the mobile medical platform. Mobile terminals can collect users’ information in a wide range of places and over a long period. Informing users more clearly and straightforwardly about what information will be collected and how it will be used can increase their trust. Operators can provide more information about doctors, such as popularity, professional level, educational level, awards, and qualifications of doctors, to increase users’ sense of security.

As the weight of the “Service” dimension increases, users have higher requirements for services. The “Service” dimension is the user’s evaluation of the platform’s customer service and the attitude of doctors. Negative comments on the “Service” dimension focused on two aspects: the time it takes to obtain a refund and the length and speed of the doctor’s response. Due to Q&A time constraints, physicians may not have enough time to provide high-quality answers in time. Some users measure doctors’ attitudes by their speed of reply and the length of their answers to questions. Users lack the professional knowledge to judge whether a doctor is providing high-quality consultation services. Therefore, most users judge the attitude of doctors in answering questions. Steven et al. found that high-quality customer service improves customer satisfaction [59]. It is recommended to introduce a doctor consultation evaluation system to help users evaluate doctors from various aspects and improve user satisfaction.

Although users pay less attention to the dimension of “Bug report”, they have the lowest satisfaction with the attribute of ”Bug report”. The weight of the “Bug report” dimension increased in 2019 and decreased in 2020, which may be related to the number of newly registered users and the change in usage. After the outbreak of COVID-19, the company’s platform visits reached 1.11 billion, the number of newly registered users increased by 10 times, and the average daily consultation volume reached 9 times that of ordinary users [48,50]. The Ping An Health App can provide users with special channels for feedback problems such as software flashback and mismatch, and appropriately provide some rewards (such as points) to encourage users to give feedback through special channels to reduce bad reviews from such users.

Hong Lin divided the Ping An Health App into 4 functional features and 26 theme features and pointed out that the satisfaction of interface and attitude was the highest, while the satisfaction of stalling and unloading was the lowest [60]. Chen Ting proposed six satisfaction indicators: doctor’s care, perceived doctor’s professionalism, perceived ease of use, perceived usefulness, system reliability, and system functional characteristics. Doctor care and doctor professionalism were significantly correlated with satisfaction with mobile medical apps [61]. Compared with the two researchers, this study combines the characteristics of application software in the process of theme mining and dimension mining and uses comment data instead of questionnaire data in the process of weight calculation. At the same time, this study explores the influence degree of dimensions and the changing trend of satisfaction in the past five years. The results show that the influence degree of each dimension on user satisfaction is not invariable.

## 5. Conclusions

With the development of mobile health, more and more users are using m-Health apps. This paper takes the Ping An Health App reviews as the research object. First, the topic model is used to mine the review topic and summarize the dimensions included in the review. Then, the sentiment analysis is used to obtain the dimension and dimension emotion of each review, calculate the weight of each dimension and user satisfaction, and finally obtain the changing trend of the weight of each dimension and user satisfaction of each dimension. Generally speaking, the key factors affecting user satisfaction are ”Feature information”, ”Usage”, ”Reliability”, ”Other”, ”Service”, ”Bug report”, and ”Background”. According to the data over the most recent 5 years, the three dimensions of ”Feature information”, ”Reliability”, and ”Service” have the greatest impact on online reviews of apps. Among them, users pay the most attention to the ”Feature information” dimension, while the ”Reliability” dimension receives increasing attention, but users have low satisfaction with these two dimensions. User satisfaction with the Ping An Health App depends on whether the user is satisfied with its core functionality. In recent years, with the increase in users’ use time, users’ awareness of privacy protection is gradually enhanced, and they pay more attention to the reliability and professionalism of mobile applications than in the past. The satisfaction of the “Bug report” dimension is the lowest, and when a user comments about the “Bug report” dimension, they are negative.

According to the results of this study, the Ping An Health. App operators can improve user satisfaction from three dimensions: “Feature information”, “Reliability” and “Service”. The Ping An Health App Software developers can improve the software interaction, enhance the stability of the system and improve the quality of the software based on the feature words extracted from the dimensions of “Usage” and “Bug report”.

Our study is subject to the validity of internal threats and external threats. Taking into account that the data we obtain is not necessarily complete, and some meaningless comments may be ignored in the process of data cleaning; manual labeling behavior during sentiment analysis can also lead to bias. This can lead to potential bias and unreasonable conclusions.

This study also has some limitations: it only selects the review data from the Apple App Store without considering the data from other platforms and other forms. In future research, the sample size will be further expanded. For example, in future research, the reviews of the Ping An Health App global edition will be compared with that of the Ping An Health App China edition to explore the differences in the dimensions that users pay attention to and the changing trend of user satisfaction in various countries, and the analysis can also be combined with cross-cultural theory. the differences in the dimensions of user attention and the changing trend of user satisfaction in different countries will be explored, and the research can also be conducted based on cross-cultural theory. In this study, the pre-trained model SKEP is used, and the amount of manually labeled data is relatively small. In future research, other pre-trained models will be tried, and the number of samples will be increased to improve the accuracy of dimensional sentiment analysis. This paper does not explain the reasons for the changes in influencing factors, such as why the weight of the “Bug report” dimension increased and then decreased in 2020, and what factors caused this change, which can be further studied in the future. Further data comparison can be obtained in future research, such as the study of users’ review intention by linking the time when users download the App with the time when users review and the sentiment of the reviews. For example, the time when users download the Ping An Health App is compared with the time when users make comments on the ”bug report” dimension, and the questionnaire survey is combined to explore why the user satisfaction with the ”bug report” dimension is the lowest.

## Figures and Tables

**Figure 1 ijerph-20-01591-f001:**
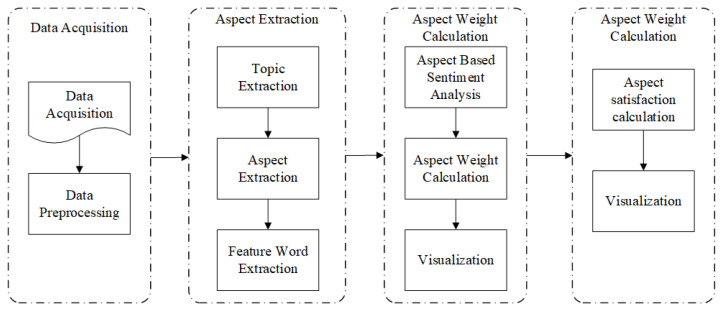
Research Framework.

**Figure 2 ijerph-20-01591-f002:**
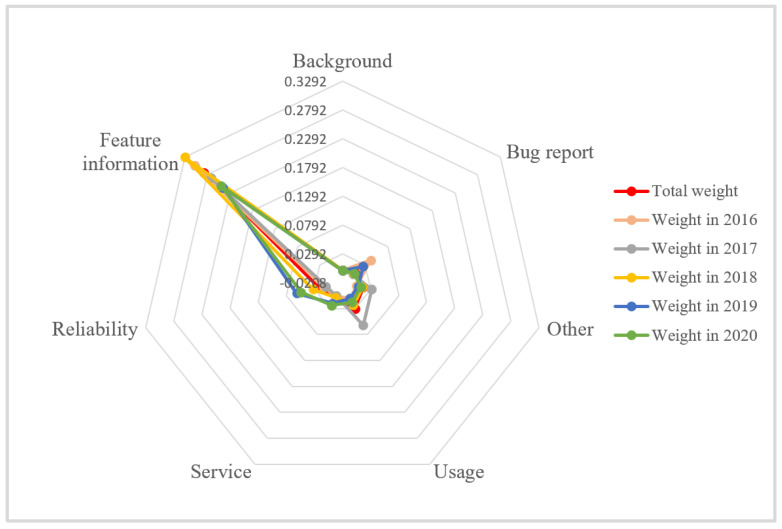
2016–2020 Ping An Health App Weight Radar Chart.

**Figure 3 ijerph-20-01591-f003:**
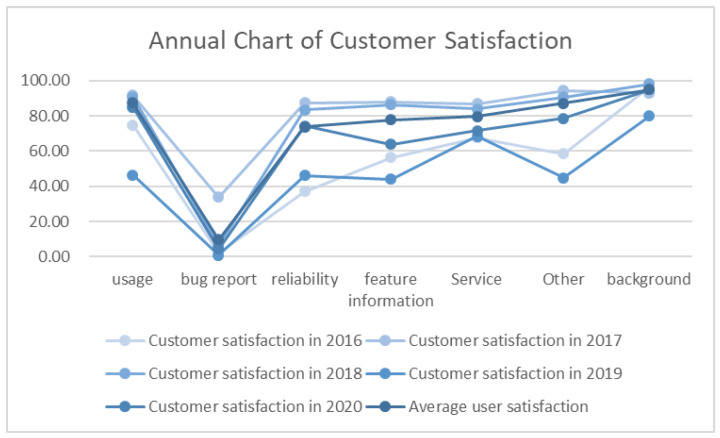
Annual Chart of Customer Satisfaction of the Ping An Health App.

**Table 1 ijerph-20-01591-t001:** Dimensions and Feature Words Table.

Dimension	Sub-Dimension	Feature Words
Usage	Usability	simple, complex, stable
	UI	fresh, beautiful, page, interface
Bug report	Bug or Crash report	flashback, stuck, crash
	Device	Android, Apple, Mobile, Ipad
	OS Version	upgrade, version, new version, update
Reliability	Security	account, login, logout
	Privacy	privacy, confidentiality
	Qualification of a physician	experts, certification, effect, quack
	News Feed	SMS, advertisement, message
Feature Information	Medical Consultation	consultation, registration, inquiry, department
	Shopping	medicine, pharmacy, delivery, Taobao
	Information Acquisition	live broadcast, weight loss, knowledge, health preservation
	Interactive Activities	health circle, prizes, promotion, gifts, member points
	Medical Expense	free, charge, recharge
Service	Service attitude	after-sales, service, attitude
Other Apps	Reference to Other App for Comparison	DXY, ledongli, 360kad, Chunyu Doctor, type, recommendation
User Types and Scenarios	User Types and Scenarios	Young people, middle-aged people, elders, white-collar workers, queuing

**Table 2 ijerph-20-01591-t002:** Dimension weight table of Ping An Health App.

Dimension	Mentioned Probability	Consistency of Sentiment	Weight
Feature information	0.4584	0.6246	0.2863
Usage	0.1458	0.1976	0.0288
Reliability	0.1194	0.1680	0.0201
Other	0.1051	0.1490	0.0157
Service	0.0743	0.1063	0.0079
Bug report	0.0781	0.0886	0.0069
Background	0.0189	0.0269	0.0005

**Table 3 ijerph-20-01591-t003:** Ranking Table of dimension weight of the Ping An Health App from 2016 to 2020.

Rank	2016	2017	2018	2019	2020	Total
1	Feature information	Feature information	Feature information	Feature information	Feature information	Feature information
2	Bug report	Usage	Reliability	Reliability	Reliability	Usage
3	Reliability	Other	Usage	Bug report	Service	Reliability
4	Usage	Reliability	Other	Service	Usage	Other
5	Service	Service	Service	Usage	Other	Service
6	Other	Bug report	Bug report	Other	Bug report	Bug report
7	Background	Background	Background	Background	Background	Background

**Table 4 ijerph-20-01591-t004:** User satisfaction scores of Ping An Health App in various dimensions from 2016 to 2020.

Year	Usage	Bug Report	Reliability	Feature Information	Service	Other	Background
2016	74.90	1.88	37.14	56.29	67.66	58.54	96.15
2017	91.73	33.93	87.43	87.97	86.97	94.24	93.02
2018	90.93	6.33	83.44	86.37	83.90	90.53	98.08
2019	46.27	0.88	46.15	43.90	68.42	44.83	80.00
2020	85.05	4.46	74.33	63.86	71.56	78.62	94.87
Total	87.37	9.66	73.60	77.68	79.69	86.99	94.75

**Table 5 ijerph-20-01591-t005:** Ranking Table of User satisfaction scores of Ping An Health App from 2016 to 2020.

Rank	2016	2017	2018	2019	2020	Total
1	Background	Other	Background	Background	Background	Background
2	Usage	Background	Usage	Service	Usage	Usage
3	Service	Usage	Other	Usage	Other	Other
4	Other	Feature information	Feature information	Reliability	Reliability	Service
5	Feature information	Reliability	Service	Other	Service	Feature information
6	Reliability	Service	Reliability	Feature information	Feature information	Reliability
7	Bug report	Bug report	Bug report	Bug report	Bug report	Bug report

## Data Availability

The data presented in this study are available upon request from the corresponding author.

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
