# Peer review of "A Study of the Ping An Health App Based on User Reviews with Sentiment Analysis"

_ijerph, 2023, doi:10.3390/ijerph20021591_

Round 1
Reviewer 1 Report
Considering that the manuscript applies a methodology to evaluate results of a mobile health app application, the authors should highlight the original contributions of the applied methodology used in this work. The quality of the paper can be improved by extending the comparative discussion.
Author Response
Dear Editor and Reviewers,
Thank you for your letter and for the reviewers’ comments concerning our manuscript entitled “A Study of the Ping An Good Doctor App Based on User Reviews with Sentiment Analysis” (ID: ijerph-2116266). Those comments are all valuable and very helpful for revising and improving our paper, as well as the important guiding significance to our research. We have studied the comments carefully and have made a correction which we hope meets with approval. The main corrections in the paper and the responses to the reviewer’s comments are as flowing:
Point 1: Considering that the manuscript applies a methodology to evaluate results of a mobile health app application, the authors should highlight the original contributions of the applied methodology used in this work. The quality of the paper can be improved by extending the comparative discussion.
Response 1:
“Hong Ling divided Ping An Good Doctor into 4 functional features and 26 theme features and pointed out that the satisfaction of interface and attitude was the highest, while the satisfaction of stalling and unloading was the lowest [58]. Chen Ting proposed six satisfaction indicators: doctor's care, perceived doctor's professionalism, perceived ease of use, perceived usefulness, system reliability, and system functional characteristics. Doctor care and doctor professionalism were significantly correlated with satisfaction with mobile medical apps[59]. Compared with the two researchers, this study combines the characteristics of application software in the process of theme mining and dimension mining and uses comment data instead of questionnaire data in the process of weight calculation. At the same time, this study explores the influence degree of dimensions and the changing trend of satisfaction in the past five years. The results show that the influence degree of each dimension on user satisfaction is not invariable.”
Special thanks to you for your good comments.
Reviewer 2 Report
Thank you for the opportunity to review the article “A Study of the Ping An Health App Based on User Reviews with Sentiment Analysis”. The paper addresses an interesting and well researched theme in the recent period about the implications of mobile health and medical applications used by patients or other interested individuals and the user satisfaction with mobile medical apps. The paper is also in line with the section “Digital Health” that was submitted to.
This study represents a solid effort in the field approached. It is constructed in a mature manner, following the publication standards of the journal, discussing the subject that needs to be comprehensively analyzed because “the number of m-Health app users is increasing, and some users are willing to give feedback on m-Health apps through the app store. User reviews may include users’ experience and functional requirements for apps, which can help software developers understand users’ attitudes and improve software quality”, as the authors underline.
Also, the study is written in an adequate manner, with a specific review of the literature and robust research design. The results are presented clearly and coherently, using visuals and text. The figures presented in the paper were relevant to explore the results of the research and the ways these were adapted to the explanations in the text.
Moreover, there are some point-by-point observations that should be addressed in this revision.
- Line 7 “Apple APP Store” replace with “Apple App Store” and replace APP with App/ app in several other places in the text
- Line 17: The first citation should be updated with more relevant and up to date technology comparison, not PDAs.
- Line 18: “The rise of Web 2.0 and Web 3.0…” citation needed
- Line 19-22: What is the reason to present the statistics from 2010 and 2017 and not have the numbers for 2021 or 2022? https://www.statista.com/ with give a quick answer to that or other relevant paper cited
- Line 50: the sentiment analysis is “fine-grained”, please use a reference to explain the procedure based on deep learning that provides an end-to-end analysis by transforming aspects of objects extraction and sentiment classification problems into sequence labelling problems that are then solved by use of deep neural networks (Li et al., 2020)
- Line 56 “App Store” replace with “App Stores”
- Line 58 “for windows phones” replace with “for “Windows phones”
- Please specify more clearly the objective(s) of the study, present the research questions (if any)
- Line 104 “This study uses the reviews of the Apple App Store”. Why only Apple App Store? The idea is presented at the end of the paper, about the Limitations of the study but the decision should be described in the Methodology section, as well.
- For Figure 2, the font should be larger
- The “Ping An Good Doctor” is not described, please add some details, features, services, etc. to the global reader and why is so important the analysis for this app in regards to other apps. Other external sources present the case: https://sph.nus.edu.sg/wp-content/uploads/2021/11/Good-Dr-Case-Study.pdf.
- Future directions of the research should be added and also to highlight the novelty factor better.
Author Response
Dear Editor and Reviewers,
Thank you for your letter and for the reviewers’ comments concerning our manuscript entitled “A Study of the Ping An Good Doctor App Based on User Reviews with Sentiment Analysis” (ID: ijerph-2116266). Those comments are all valuable and very helpful for revising and improving our paper, as well as the important guiding significance to our research. We have studied the comments carefully and have made a correction which we hope meets with approval. The main corrections in the paper and the responses to the reviewer’s comments are as flowing:
- Line 7 “Apple APP Store” replace with “Apple App Store” and replace APP with App/ app in several other places in the text
Line 7: “Apple APP Store” was corrected as “Apple App Store”
- Line 17: The first citation should be updated with more relevant and up to date technology comparison, not PDAs.
Line 17: Considering the Reviewer’s suggestion, we have changed this into “Mobile health, or m-health, is a promising tool for supporting healthcare, using mobile technologies and innovative applications to solve health problems that can provide information and resources to healthcare professionals and patients.”
- Line 18: “The rise of Web 2.0 and Web 3.0…” citation needed
Line 18: “The rise of Web 2.0 and Web 3.0…” citation was added
- Line 19-22: What is the reason to present the statistics from 2010 and 2017 and not have the numbers for 2021 or 2022? https://www.statista.com/ with give a quick answer to that or other relevant paper cited
Line 19-22: As Reviewer suggested that we re-provide the statistics for 2010 and 2022.
- Line 50: the sentiment analysis is “fine-grained”, please use a reference to explain the procedure based on deep learning that provides an end-to-end analysis by transforming aspects of objects extraction and sentiment classification problems into sequence labelling problems that are then solved by use of deep neural networks (Li et al., 2020)
Line 50: We have rewritten this part according to the Reviewer’s suggestion.
- Line 56 “App Store” replace with “App Stores”
Line 56: “App Store” was corrected as “App Stores”
- Line 58: “for windows phones” replace with “for “Windows phones”
Line 58: “for windows phones” were corrected as “for Windows phones”
- Please specify more clearly the objective(s) of the study, present the research questions (if any)
Line 90: the research questions were added
- Line 104 “This study uses the reviews of the Apple App Store”. Why only Apple App Store? The idea is presented at the end of the paper, about the Limitations of the study but the decision should be described in the Methodology section, as well.
Line 104:the decision “Because Ping An Good Doctor app offers a global edition on the Apple App Store, this study uses the reviews of the Apple App Store” was added.
- For Figure 2, the font should be larger
As Reviewer suggested that we have changed the font in Figure 2.
- The “Ping An Good Doctor” is not described, please add some details, features, services, etc. to the global reader and why is so important the analysis for this app in regards to other apps. Other external sources present the case: https://sph.nus.edu.sg/wp-content/uploads/2021/11/Good-Dr-Case-Study.pdf.
As Reviewer suggested that the description of “Ping An Good Doctor” was added. Reference 47 illustrates this case
Line 184-188: “As the largest national online medical portal in China, Ping An Good Doctor is a widely used online medical service platform, aiming to build a one-stop, full-process, online-to-offline service platform, and an online comprehensive health service platform integrating private clinics, pharmacies, health check-up, testing centers, and other offline health services\cite{ref-journal32,ref-url3}. By the end of 2019, Ping An Good Doctor had 315 million registered users and 729,000 daily consultations for medical services throughout the year\cite{ref-url4}.”
- Future directions of the research should be added and also to highlight the novelty factor better.
We have rewritten this part according to the Reviewer’s suggestion.
Line 375-395: “This study also has some limitations: it only selects the review data from the Apple App Store without considering the data from other platforms and other forms. In future research, the sample size will be further expanded. For example, in future research, the reviews of the Ping An Good Doctor App global edition will be compared with that of the Ping An Good Doctor App China edition to explore the differences in the dimensions that users pay attention to and the changing trend of user satisfaction in various countries, and the analysis can also be combined with cross-cultural theory. the differences in the dimensions of user attention and the changing trend of user satisfaction in different countries will be explored, and the research can also be conducted based on cross-cultural theory. In this study, the pre-trained model SKEP is used, and the amount of manually labeled data is relatively small. In future research, other pre-trained models will be tried, and the number of samples will be increased to improve the accuracy of dimensional sentiment analysis. This paper does not explain the reasons for the changes in influencing factors, such as why the weight of the "Bug report" dimension increased and then decreased in 2020, and what factors caused this change, which can be further studied in the future. More data comparison can be obtained in future research, such as the study of users' review intention by linking the time when users download the App with the time when users review and the sentiment of the reviews; For example, the time when users download the Ping An Good Doctor App is compared with the time when users make comments on the bug report dimension, and the questionnaire survey is combined to explore why the user satisfaction of the bug report dimension is the lowest.”
Special thanks to you for your good comments.
Reviewer 3 Report
This is an interesting paper dealing with the evaluation of mHealth apps based on sentiment analysis. A couple of issues
a) Has COVID affected the user outcomes?
b) Bug solution has the worst user rating. How is this explained with the higher reliability meaures?
c) In the methods please add information on the architecture and fumction of the NLP DL tools used and discuss possible biasing.
Author Response
Dear Editor and Reviewers,
Thank you for your letter and for the reviewers’ comments concerning our manuscript entitled “A Study of the Ping An Good Doctor App Based on User Reviews with Sentiment Analysis” (ID: ijerph-2116266). Those comments are all valuable and very helpful for revising and improving our paper, as well as the important guiding significance to our research. We have studied the comments carefully and have made a correction which we hope meets with approval. The main corrections in the paper and the responses to the reviewer’s comments are as flowing:
Point 1:Has COVID affected the user outcomes?
Response 1:Two references pointed out that the average daily consultations of Ping An Good Doctor increased after the COVID-19 outbreak. We added a discussion about bug report dimensions.
Line 336-345.” Although users pay less attention to the dimension of "Bug report", they have the lowest satisfaction with the attribute of Bug report. The weight of the "Bug report" dimension increased in 2019 and decreased in 2020, which may be related to the number of newly registered users and the change in usage. After the outbreak of COVID-19, the company's platform visits reached 1.11 billion, the number of newly registered users increased by 10 times, and the average daily consultation volume reached 9 times that of ordinary users [45, 47]. Ping An Good Doctor App can provide users with special channels for feedback problems such as software flashback and mismatch, and appropriately provide some rewards (such as points) to encourage users to give feedback through special channels to reduce bad reviews from such users.”
Point 2:Bug solution has the worst user rating. How is this explained with the higher reliability meaures?
Response 2:In future research, we hope to add the time when users download the App, compare it with the time when users make comments on the bug report dimension, and explore why the bug report dimension has the lowest user satisfaction combined with the questionnaire survey.
Line 389-395: “More data comparison can be obtained in future research, such as the study of users' review intention by linking the time when users download the App with the time when users review and the sentiment of the reviews; For example, the time when users download the Ping An Good Doctor app is compared with the time when users make comments on the bug report dimension, and the questionnaire survey is combined to explore why the user satisfaction of the bug report dimension is the lowest..”
Point 3:In the methods please add information on the architecture and function of the NLP DL tools used and discuss possible biasing.
Response 2:We have made corrections according to the Reviewer’s comments, the information and function of the NLP DL tools used and possible biasing were added.
Line 160-175:” In this paper, a deep semantic sentiment analysis framework, PaddleNLP, is used to conduct fine-grained sentiment analysis on review text. PaddleNLP is an easy-to-use and powerful NLP library with an awesome pre-trained model zoo, supporting a wide range of NLP tasks from research to industrial applications [44]. PaddleNLP supports several pre-trained models, its Sentiment_Analysis module is based on SKEP pre-trained model, PP-MiniL pre-trained model, and quantized PP-MiniL pre-trained model[45] . The whole process of sentiment analysis roughly includes two stages, which are the review opinion extraction model and the attribute level sentiment classification model. For a given piece of text, we first extract the potential comment attribute and the corresponding opinion of the attribute in the text sentence based on the former, and then concatenate the comment attribute, opinion, and the original text, and pass it to the aspect-level sentiment classification model to identify the sentiment polarity of the review attribute.”
Line 370-374: “Our study is subject to the validity of internal threats and external threats. Taking into account that the data we obtain is not necessarily complete, and some meaningless comments may be ignored in the process of data cleaning; manual labeling behavior during sentiment analysis can also lead to bias. This can lead to potential bias and unreasonable conclusions.”
Special thanks to you for your good comments.
Round 2
Reviewer 2 Report
Thank you for the opportunity to review the revised version of the paper “A Study of the Ping An Health App Based on User Reviews with Sentiment Analysis”.
The authors responded and explained with reasonable arguments and corrected all the remarks and observations highlighted in the previous review and the results suggest a more consistent and logical text.
To sum it up, the authors developed a more in-depth theoretical presentation about the subject, integrating the suggested aspects of the review.
I consider that the paper is publishable after a final check from the authors, especially correcting the References by using the standards and guidelines of the IJERPH journal.
One final remark for the authors: the new manuscript version of the modified text should have been marked with different color to identify more quickly the changes made in the first version.